# To Screen or Not to Screen: “False Positive” Cases—Can They Be Treated as Definitely False? Properly Selecting the Screening Age-Range Groups in Scoliosis Screening Programs

**DOI:** 10.3390/healthcare13060600

**Published:** 2025-03-10

**Authors:** Theodoros B. Grivas, Elias Vasiliadis, Christina Mazioti, Aristea Mamzeri, Despina Papagianni, Galateia Katzouraki, Nikolaos Sekouris

**Affiliations:** 1Department of Orthopedics & Traumatology, “Tzaneio” General Hospital of Piraeus, 185 36 Piraeus, Greece; 23rd Department of Orthopaedics, School of Medicine, National and Kapodistrian University of Athens, KAT Hospital, 145 61 Athens, Greece; eliasvasiliadis@yahoo.gr; 3“Tzaneio” General Hospital of Piraeus, 185 36 Piraeus, Greece; maziotix@gmail.com; 4TOMY Attica Square, 104 45 Athens, Greece; mamzeri_aristea@hotmail.com; 5Special Primary School of Rafina, 190 09 Rafina, Greece; papdes2009@hotmail.com; 6Spinal Department of Hygeia Hospital, 4 Erythrou Stavrou, 151 23 Maroussi, Greece; gkatzouraki@hotmail.com; 7The 1st Department of Orthopedics, P. & A. Kyriakou Children’s Hospital, 23 Levadeias, 115 27 Athens, Greece; nick_sekouris@yahoo.com

**Keywords:** idiopathic scoliosis, school scoliosis screening, false positive cases, scoliosis screening age range, school scoliosis screening

## Abstract

**Background/Objectives:** This opinion paper provides a brief overview of the history of school scoliosis screening programs following the introduction. **Methods:** It outlines the international administrative policies of these programs, their impact on the frequency of surgical procedures, and the effects of discontinuing school-based scoliosis screenings. **Results:** The primary focus is on analyzing the role of “false positive” cases detected during the Adam’s bending test, which has contributed to the discontinuation of these programs in certain countries. This focused discussion is based on the impact of growth on the relationship between spinal and rib cage deformities. Furthermore, we propose the selection of the optimal age range for screening, considering the correlation between idiopathic scoliosis prevalence and geographical latitude. **Conclusions:** Lastly, we emphasize the importance of continuing scoliosis screening programs in schools.

## 1. Introduction

Idiopathic scoliosis (IS) is a three-dimensional deformity of the spine and thorax. In reality, it is a four-dimensional condition, with the fourth dimension being time—scoliosis persists throughout a person’s lifetime. The exact cause of IS remains unclear [1]. Various concepts and hypotheses have been proposed regarding its development, drawing from studies on genetics, the central nervous system (CNS), spinal growth and bone metabolism, metabolic pathways, biomechanics, and other factors. However, these studies primarily explain the mechanisms underlying scoliosis rather than its etiology [2,3,4].

The prevalence of IS varies significantly across studies and countries, ranging from 0.93% to 12% [5]. This variation can be attributed to various biological and geographical factors [1]. However, several publications report a narrower prevalence range in the general population [6,7,8,9,10,11]. These studies do not provide a comprehensive analysis of the full spectrum of IS prevalence or the reasons behind its variability, except for a limited number of publications, such as Wong et al. (2005) [12]. Wong et al. reported that prevalence rates were low at ages 6 to 7 and 9 to 10 but increased rapidly to 1.37% and 2.22% among girls aged 11 to 12 and 13 to 14, respectively. They also found that the false-positive rate in girls decreases with age, likely due to the concurrent increase in scoliosis prevalence [12].

It has been reported that the age at menarche is associated with the prevalence of idiopathic scoliosis (IS) [13]. A later age at menarche corresponds to a higher prevalence of adolescent idiopathic scoliosis (AIS). Moreover, IS prevalence is thought to be influenced by geographic latitude and sunlight exposure, which affect the age at menarche and, consequently, the development of AIS. Additionally, the age at menarche varies across countries, which may partially explain the wide variation in reported IS prevalence worldwide [13].

The treatment of IS varies based on curve severity, age at presentation, and growth potential [14,15,16,17,18]. Management begins with prevention, primarily through early recognition of the deformity via school scoliosis screening (SSS). The primary goal of modern SSS programs is to identify individuals with truncal asymmetry (TA) and/or unrecognized IS at an early stage when less invasive treatment is most effective. Additionally, the screening process has been shown to reduce morbidity within the population at a negligible cost [19].

For mild to moderate scoliotic curves, physiotherapeutic-specific scoliosis exercises (PSSE), bracing, or a combination of both non-operative treatments aim to prevent progression or even improve the deformity to some extent [20]. In cases of severe IS, surgical intervention is required, utilizing various techniques and approaches, particularly for skeletally immature patients [21,22,23].

## 2. The History of School Scoliosis Screening (SSS)

The preventive screening for scoliosis as part of general health assessment programs for young people, particularly school children, began in Europe at the start of the 20th century. In Greece, school doctors also examined children for scoliosis. Evidence suggests that such programs were implemented in various European countries and cities, as documented in the booklet H ΣKOΛIΩΣIΣ, written by Dr. Em. N. Lambadarios and published in Athens in 1915 by Thanos Tzavelas (Figure 1). The recorded locations included Hamburg, Lausanne, Paris, Athens, and Sweden.

Fifty years later, in 1963, the SSS programs was initiated in Aitken, Minnesota, under the leadership of Dr. Lonstein. He emphasized the significant benefits of early screening, stating that detecting spinal curves in their early stages allows for timely intervention, including bracing, which helps prevent further progression and reduces the likelihood of future surgical correction. Routine scoliosis screening in Minnesota schools has proven to be an effective method for the early detection of spinal deformities. Regarding the SSS program, Dr. Lonstein described the screening process as “a rapid and easy test—a 30 s investment for a lifetime of dividends” [24,25,26].

Dr. Dean MacEwen played a key role in the early development of voluntary SSS programs by implementing them in Delaware schools during the 1960s [27].

In 1975, Dr. Panagiotis Smyrnis organized and conducted the first SSS program in Athens, Greece. A total of 3494 children aged 12–14 from the Attica region were visually examined using the Adams forward bending test. The study found that 10% of the children tested positive clinically, while radiologically confirmed scoliosis (curves of 10 degrees or more) was observed in 4.6% of girls and 1.1% of boys. Additionally, the study noted that children with blond hair and blue eyes were more prone to scoliosis compared to those with darker features.

Dr. Smyrnis emphasized that the bending test remains the most commonly used and accessible screening method due to its simplicity, even for non-experts. He also highlighted the importance of a second evaluation before proceeding with radiological examination, as well as the benefit of conducting screenings in a comfortable and familiar setting. He stated, “It should be emphasized that apart from the size of the hump, its spinal level has clinical significance”. Furthermore, he recommended that mass screening should focus on children aged 10–13, as 85% of clinically significant idiopathic scoliosis (IS) cases emerge during this period. The study also observed a 3:1 ratio between clinically and radiologically positive cases [28,29].

The results of this program were first presented at the third “N. Giannestras” Symposium in September 1975 in Athens, Greece [29,30], and later at the sixth Symposium held at the Cardio-thoracic Institute, Brompton Hospital, London, on 17–18 September 1979 [31].

Consequently, similar SSS programs were implemented by orthopedic teams interested in conducting SSS in various Hellenic cities. The SSS program at “Thriasio” Hospital was conducted from 1996 to 2009, followed by the “Tzaneio” Hospital program from 2009 to 2019. During this period, TBG served the Hellenic National Health System, examining a total of 24,223 children. Further details on the history and reported outcomes of Hellenic SSS programs up to the year 2000 can be found in the book School Screening in Greece [28].

## 3. Policies of Administration of SSS Programs

Global policies on the implementation of SSS programs exhibit both encouraging and discouraging trends, reflecting inconsistency in approach [32,33]. The necessity of an SSS program for IS was initially debated [34,35,36,37].

In the Netherlands, Pruijs et al. (1996) reported that budget cuts had led to a decline in the frequency of health checks in many districts compared to the 1980s. These checks included the forward-bending test, a key component of scoliosis screening [38].

For years, the U.S. Preventive Services Task Force (USPSTF) maintained a negative stance on SSS implementation. In 1996, it found insufficient evidence to support or oppose routine screening for asymptomatic adolescents with IS, issuing an “I” recommendation (indicating insufficient evidence) [39]. By 2004, the USPSTF shifted its position, recommending against routine screening for asymptomatic adolescents [40,41,42].

This change was met with concern from major medical societies, including the American Academy of Orthopaedic Surgeons (AAOS), the Scoliosis Research Society (SRS), the American Academy of Pediatrics (AAP), and the Pediatric Orthopaedic Society of North America (POSNA). These societies criticized the USPSTF’s updated recommendation, emphasizing that it was made without substantial changes in the existing literature, without revised position statements from key medical societies, and without significant input from specialists who regularly treat children with IS. The AAOS, SRS, POSNA, and AAP did not endorse any formal recommendations against SSS based on the available evidence.

In its most recent report, the USPSTF acknowledged that screening can detect AIS. Furthermore, it recognized that bracing and possibly exercise-based treatments can slow or prevent the progression of spinal curvature during adolescence [43].

In 2007, the topic of SSS was addressed at the SOSORT meeting, leading to a consensus paper that aimed to summarize the current state of SSS and outline areas of agreement among 35 health professionals worldwide. The SOSORT endorsed SSS programs [33].

In 2010, during the SRS meeting in Kyoto, the validity of SSS programs was questioned. The SRS presidential line assigned eight international experts to investigate SSS from a global perspective. The SRS International Task Force on Scoliosis Screening examined the clinical effectiveness, technical program effectiveness, cost-effectiveness, and treatment outcomes. Their final conclusion was that SSS programs should be recommended [33].

The implementation of SSS is closely tied to non-operative treatment for Idiopathic Scoliosis (IS). At the 48th annual SRS meeting in Lyon, the notable BRAIST study was presented. Dr. Weinstein, the presenter, reported that bracing adolescents with moderate IS effectively reduced the number of patients requiring surgery. The study also showed a positive correlation between the number of hours spent wearing the brace and its success. The findings suggested that early detection, brace compliance, improved bracing indications, cost savings from bracing versus surgery, and a reconsideration of screening policies were crucial [44,45,46,47,48,49].

Recently, there has been a global trend in favor of implementing SSS programs, including endorsement from the USPSTF. It seems that the momentum is shifting back toward scoliosis screening [50].

Our training center at the “Tzaneio” hospital in Piraeus, which trains certified SSS program examiners, was officially recognized by the Hellenic State (Government Gazette No. Sheet 4168, Issue B, 30 November 2017) as the “Special Training Center for the Implementation of the Student Screening Program for Scoliosis and Other Spinal Deformities”.

In 2012, Plaszewski et al. analyzed various countries’ approaches to SSS based on publications up until that year [51].

In North America, fewer than half of the U.S. states have legislated SSS programs, and there is no nationwide requirement or standard for screening [14,25,52,53,54]. Canada introduced SSS programs in the 1970s, but these were gradually stopped. A recent article highlights that “Canada no longer has regular childhood screening for scoliosis, and other forms of prevention are not receiving the attention that surgery does. Some doctors are advocating for change” (Oct 2023) [55].

In Australia, SSS was introduced intermittently, but by the early 1990s, most programs were abandoned, with medical societies now advocating for a National Self-Detection Program for Scoliosis [56].

In Asia, countries like India [57], Saudi Arabia [58], Hong Kong, and parts of China have active SSS programs, with Hong Kong incorporating it into their Student Health Service since 1995 [59,60,61,62,63]. Singapore has included routine SSS in its national school health program since 1981 [12,64]. Other Southeast Asian countries, such as Malaysia [65,66], Indonesia [67], Vietnam [68], and Korea [69,70], occasionally carry out localized SSS programs. Japan has a mandatory SSS program, though it is organized locally [71].

In Africa, Nigeria has implemented an SSS program [72].

In South America, some Brazilian cities occasionally implement SSS programs [73,74], while Chile has carried out an SSS program in its metropolitan region [75].

In Europe, policies and legislation regarding SSS vary. The UK and Poland do not have a national policy, while countries like Sweden [76], Spain [6], the Netherlands [38], Italy [29], Greece [77,78,79,80,81], Bulgaria [82], Turkey [83,84], and Israel [85] offer SSS on a voluntary basis.

## 4. The Impact of Screening on Frequency of Surgical Treatment

The previous data on this issue seem somewhat inconsistent and inconclusive. For instance, in Minnesota, USA, where SSS is implemented, a decline in the frequency of IS surgeries was observed from 1974 to 1979 [25]. Torell et al. (1981) reported that SSS led to a reduction in the number of IS patients requiring surgery [86]. In another study [53], data were provided on the frequency of surgical treatment per thousand children screened over seven or more years from three US states: Kansas and Virginia showed no clear pattern, while in Minnesota, surgery rates decreased until 1981–82, after which they increased again [54]. Many European studies, however, provide stronger evidence of the positive impact of conservative treatment on reducing the frequency of IS surgeries. When high-quality conservative treatment is available, the incidence of surgery can be significantly reduced [87,88,89,90,91,92,93,94,95,96,97,98,99,100,101].

## 5. The Impact of Discontinuation of SSS

Beausejour et al. (2007) reported a concerning outlook on how the discontinuation of SSS programs impacted referral patterns for patients with AIS [102]. In a cross-sectional study conducted at a pediatric hospital’s orthopedic outpatient clinic in Canada, they assessed all patients referred for suspected AIS. The aim was to evaluate how the current referral practices compared to those before SSS programs were discontinued. Out of 489 patients referred with suspected AIS, 206 (42%) had no significant deformity (Cobb angle < 10 degrees), making these referrals inappropriate. Among patients with confirmed AIS, 91 (32%) were referred too late for proper brace treatment. The study concluded that the pause of SSS programs has resulted in suboptimal referral practices in orthopedic care for AIS [102]. Similarly, Ohrt-Nissen et al. (2016) noted that, without SSS programs, patients referred by GPs for AIS evaluation tended to have more severe curvature compared to those in systems with SSS [103]. In Norway, the frequency of brace treatment has decreased, while surgery rates have risen since the end of SSS programs, as highlighted by Adobor et al. (2012) [104]. Cho et al. (2018) found that in South Korea, patients with suspected AIS are increasingly referred late or inappropriately to tertiary centers due to the absence of SSS [105]. This outcome was widely anticipated and demonstrates how a shift toward epidemiology over early prevention and diagnosis can have detrimental effects. In a society that prioritizes people’s well-being, prevention should be a standard policy, not just a focus on statistics, epidemiology, or costs. We must recall the ancient Greek principle that “man is the measure of all things”, meaning the value of any decision or system should always center on the human being above all else.

## 6. The Reasons of SSS Discontinuation

Children with truncal asymmetry are referred to a specialized scoliosis clinic (such as those in hospitals or other medical organizations) for further assessment. The SSS team has pre-prepared and printed invitation letters for re-assessment, which are given to the school headmaster or directly to the examined children to pass on to their families.

Growth significantly affects the relationship between thoracic and spinal deformity in girls with IS. This should be considered when assessing spinal deformity using surface measurements, like a scoliometer. Research shows that, during growth, rib cage deformity occurs before spinal deformity in IS scoliogenesis [106].

The introduction of the Rib Index (RI) has provided valuable insights through statistical analysis, particularly the correlation between the Cobb angle and the RI in radiographs from referrals in our SSS programs. Approximately 30% of younger girls referred from our SSS program (under age 13) with an ATR of 7 degrees or more had either a straight spine or a Cobb angle of less than 10 degrees. In this younger age group, the correlation between clinical deformity (such as truncal asymmetry) and radiographic measurements (like the Cobb angle) is not statistically significant. However, this correlation is significant in older girls aged 14–18, as shown in Figure 2.

Growth appears to play a key role in the relationship between rib cage and spinal deformity in girls with IS. The study findings suggest that the correlation between thoracic deformity and spinal deformity is weak in younger children, indicating that rib cage deformity develops before spinal deformity in the progression of IS [106,107]. As shown in Figure 2, an RI of 2.5 corresponds to a Cobb angle of less than 10 degrees in girls aged 7–13. This indicates a notably deformed thorax [106]. Therefore, younger children identified with surface deformity using a scoliometer, but without a confirmed scoliotic curve on radiographs, are at risk for developing IS and must be monitored, not dismissed from follow-up.

This recommendation is supported by findings from Nissinen et al. (1993) in their longitudinal study [108]. Without our screening program, which included younger children alongside the typical age group, these conclusions would not have been possible. The introduction of the RI and its correlation with the Cobb angle allows us to more accurately determine the age range of children who should be monitored at the scoliosis clinic.

**Figure 2 healthcare-13-00600-f002:**
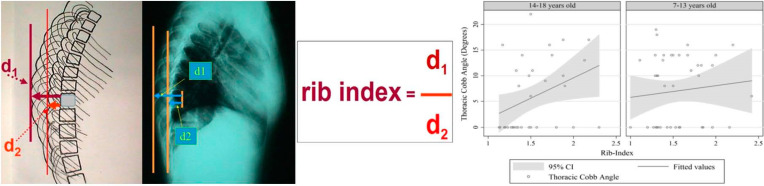
The correlation to clinical deformity (rib index) and to Cobb angle is not statistically significant in girls less than 13 yrs. old, while in older SSS referred girls aged 14–18 years old it is, (from our citation no [109,110]).

Our referral guidelines for re-examination are as follows: if the child is under 14 years old and the ATR is 5–6 degrees or more, or if the child is 14 or older and the ATR is 5 degrees or more. Children referred to the scoliosis clinic from the SSS programs undergo both clinical and radiographic evaluations, as needed.

In a lecture published in the referenced book [28], Dr. Smyrnis (2000) pointed out that the most critical age for mass screening is between 10 and 13 years old, where a 3:1 ratio is typically observed between children with clinically and radiologically positive results. The group of younger children who are referred from SSS programs for asymmetric trunk signs, but who show a straight spine or less than a 10-degree Cobb angle, are often considered “false positives” after being examined with the Adam’s bending test. However, these children may not actually be false positives, as they could develop IS as they grow, so they should not be dismissed as such. This lack of understanding, along with financial factors, likely led several medical systems to discontinue their SSS programs, as observed in the study by Yawn et al. (1999) [32]. They reported that SSS identified some children who required treatment but referred many others who did not, and this should be taken into account when making decisions about SSS. Publications like this created a negative perception of SSS, which aligns with the fact that some younger children may show trunk asymmetry without having scoliosis in their spine. The Yawn et al. study confirmed this, based on a retrospective cohort study of children who attended kindergarten or first grade in public or private schools in Rochester, MN, between 1979 and 1982.

Another reason for the unfavorable assessment of SSS value was the narrower age range of children examined in their programs, compared to our SSS program [111]. This limitation prevented them from recognizing that younger children with asymmetry during the Adam’s bending test, but no scoliosis, might later develop spinal deformities as they grow.

This more restricted age range (11–14 years old, corresponding to fifth, sixth, and seventh grades) also led to questions about the optimal cutoff point for referral to scoliosis clinics based on scoliometer readings in SSS. The “false positive” cases, where children exhibited a hump but had no spine deformity (spine straight or Cobb angle under 10 degrees), caused significant inconvenience for both doctors and parents, who invested time and money in specialized re-examinations far from their homes. Additionally, the unnecessary radiographs and the radiation exposure were seen as both costly and potentially harmful to the health of growing children. Moreover, the forward-bending test itself was criticized for its unreliability in diagnosing scoliosis [112]. Interestingly, the same study reported that among 121 spinal deformities with an initial Cobb angle less than 10 degrees, 44 (35.8%) progressed, and of 29 scoliotic deformities with initial Cobb angles between 10 and 20 degrees, 14 (48.3%) progressed, with a difference of at least 5 degrees in more than one examination [112].

## 7. Which Age Group Must Be Screened

Melatonin, “the light of night”, is secreted from the pineal gland principally at night. It is involved in sexual maturation in females [113]. Abnormally high or pharmacologic concentrations of melatonin in women are associated with altered ovarian function and anovulation. Melatonin acts in gonads indirectly, reducing the secretion of gonadotropins and mainly LH [114]. The menarche is related with episodic secretion of LH during the night [115,116]. Melatonin may play a role in the timing of puberty. The onset of puberty in humans may be related to the decline in melatonin secretion that occurs as children grow [117]. Therefore, increased melatonin levels inhibit sexual maturation in females. In northern latitude countries this sexual maturation happens later [118], due to poor light environmental conditions resulting in increased IS prevalence. It is hypothesized that the increased levels of melatonin in northern countries with poor light environmental conditions are reducing the secretion of LH and cause delayed age at menarche. Dossuse et al. 2013, reported that women born at lower latitudes or in regions with higher annual or spring/summer UVR dose had a 3–4 months earlier menarche than women born at higher latitudes or in regions with lower UVR [119].

In addition, the regression curves of prevalence of IS by latitude and age at menarche by latitude are of similar pattern [13]. Age at menarche is considered a reliable prognostic factor for IS development and progression [13,120], and varies in different geographic latitudes. A late age at menarche is parallel with a higher prevalence of AIS. This phenomenon is also observed in women with visual deficiency, and in this female group the prevalence of scoliosis is increased [121]. Pubarche appears later in girls that live in northern latitudes and thus prolongs the period of spine vulnerability, while other scoliogenic factors are contributing to the development of AIS.

A possible role of geography in the pathogenesis of IS was reported [13]. It appears that latitude which differentiates the sunlight influences melatonin secretion and modifies age at menarche, which is associated with the prevalence of IS. Age at menarche is considered a reliable prognostic factor for IS and varies in different geographic latitudes [13]. Consequently, the geographical latitude and menarche will determine the age range groups of children which will be screened. Traditionally the age group is including the fifth and sixth grade of primary school and seventh and eighth grade in high school, but this range must be adjusted according to the latitude of the place because the menarche differs at different latitudes.

## 8. Why We Must Continue School Screening Programs

It has been reported [122] that the decision not to implement screening due to cost-effectiveness concerns is based on an outdated assumption—originally derived from an early study [123] that surgery is the only proven treatment option. However, as noted by Hawes (2003), this study does not provide a scientifically justified basis for that conclusion [122].

Today, evidence demonstrates that signs and symptoms of scoliosis can be significantly improved through intensive inpatient exercise programs [124], and that the rate of progression can be substantially reduced [125]. Moreover, the incidence and prevalence of surgery can be significantly lowered when high-quality conservative treatment is available [97,98,99].

It is also well-documented and widely accepted that bracing alters the natural course of IS [86,125,126,127,128,129,130,131,132,133,134], and that SSS programs reduce the number of patients requiring surgical intervention [86].

Psychosocial health and body image concerns play a crucial role in treatment adherence and patient satisfaction, particularly among adolescents. Research suggests that psychosocial distress and body image disturbances are less pronounced in patients with strong social or family support, regular exercise habits, or good psychological well-being [135]. Additionally, SSS programs provide an opportunity to identify and document other developmental abnormalities.

Collectively, these studies support the hypothesis that school screening is justified—not only to detect and conservatively treat mild and moderate spinal curvatures but also to identify younger children with truncal asymmetry who may later develop scoliosis. Furthermore, evidence suggests that screening programs have effectively reduced morbidity at a negligible cost in regions where SSS has been implemented [19].

## 9. Conclusions

One reason SSS programs have been discontinued is the occurrence of false positive cases. Many younger children are referred from these programs due to trunk asymmetry but actually have a straight spine or a Cobb angle below 10 degrees, leading to their classification as “false positives”. However, it is highly likely that some of these children may later develop IS as they grow, meaning they should not be definitively labeled as false positives.

In areas where SSS programs have been discontinued, referrals have become less effective—either delayed or inappropriate for conservative treatment—leading to an increase in the number of surgical interventions. On the other hand, implementing SSS programs has significantly reduced the need for surgery, especially in regions where high-quality conservative treatment is available.

The recommended age group for screening is adjusted based on latitude, as the timing of menarche varies accordingly. It is advised to conduct screenings within a window of two years before and two years after the average age of menarche in a given country. Ultimately, SSS programs are strongly recommended due to their numerous benefits.

## Figures and Tables

**Figure 1 healthcare-13-00600-f001:**
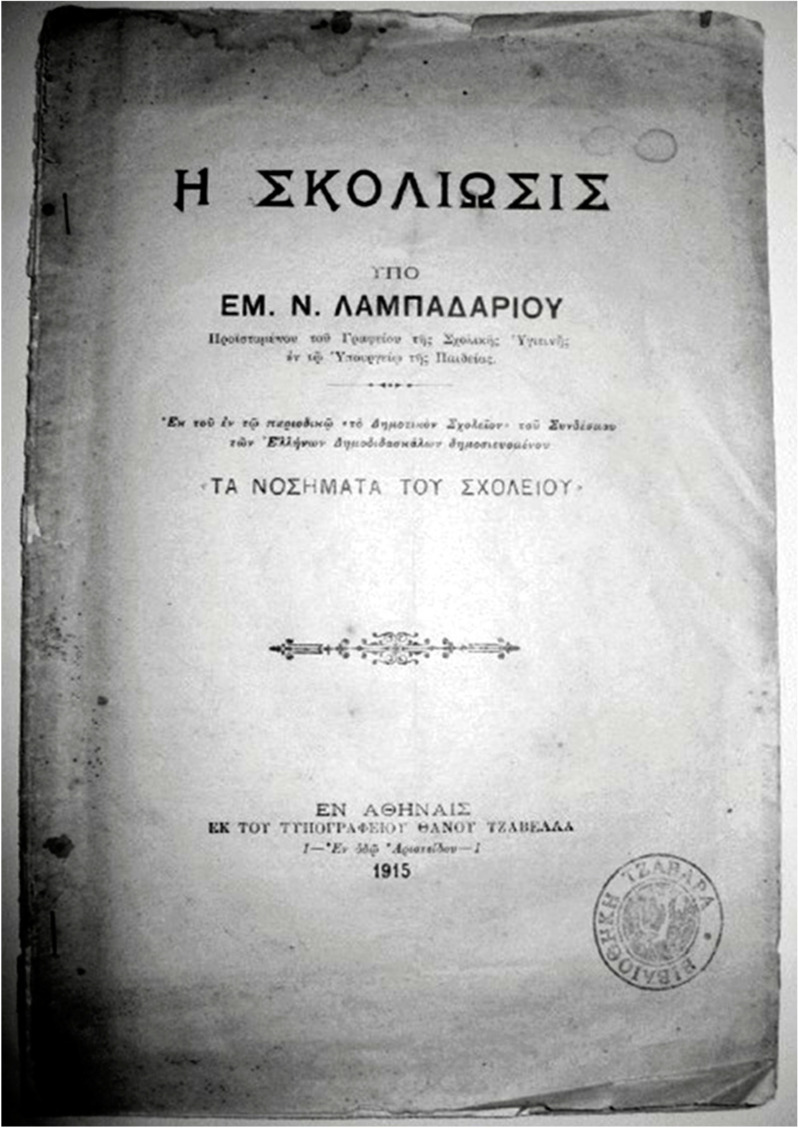
EM. N. LAMPADARIOS SCOLIOSIS. THE DISEASES OF SCHOOL. IN ATHENS. From the printing house of Thanos Tzavellas, 1 Aristidou Street, 1915. — EM. N. ΛAMΠAΔAPIOΣ H ΣKOΛIΩΣIΣ. TA NOΣHMATA TOΥ ΣXOΛEIOΥ. EN AΘHNAIΣ. Eκ του τυπογραφείου Θάνου Tζαβέλλα, Eν οδώ Aριστείδου 1, 1915.

## Data Availability

No new data were created or analyzed in this study.

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
