# Peer review of "To Screen or Not to Screen: “False Positive” Cases—Can They Be Treated as Definitely False? Properly Selecting the Screening Age-Range Groups in Scoliosis Screening Programs"

_healthcare, 2025, doi:10.3390/healthcare13060600_

Round 1
Reviewer 1 Report
Comments and Suggestions for Authors
Dear authors
According to the guidelines of the journal: Opinions are concise articles that express an author's perspective on a specific topic, method, or recent discovery. These pieces emphasize both the positive and negative aspects of the subjects discussed. While their format resembles that of a review, opinions are considerably more condensed and concentrate on the writer's personal stance rather than offering a comprehensive, analytical evaluation.
So in the reviewed article , in addition to linguistic errors and inclarity of the content, it is devoided from most of nacessary requirements of opinion articles.
Comments on the Quality of English LanguageThe english of the article need substantial revision
Author Response
Dear Reviewer 1,
I sincerely appreciate your and other editor’s time and effort in reviewing our article.
We have made revisions in accordance with your suggestions. Regarding the recommendation to improve the English language, the feedback provided was quite general and did not specify particular sections requiring refinement. Nonetheless, we have made several linguistic adjustments.
Regarding the third reviewer's comment that this article should be classified as a review rather than an opinion paper, I fully respect MDPI’s rules and guidelines.
However, I believe that every rule has exceptions, and in this case, reclassifying the article as a review paper would not be appropriate for the following reasons: The argument presented regarding the implementation of the School Scoliosis Screening (SSS) is based on 25 years of our own research, which involved extensive data collection over the course of our unique 25-year SSS program. Unlike most similar studies, our research covered a wide age range (5–18 years), allowing us to analyze and uncover significant new insights that had not been previously published. These findings challenge the rationale behind discontinuing SSS in various countries. Restricting the scope of this article by categorizing it as a review article would diminish the strength of our argument. Given the nature of the discussion and the evidence presented, it is more appropriately classified as an opinion paper.
Additionally, the paper’s subheadings on the history and administration of SSS are highly useful in presenting the international policies related to SSS.
I hope you will consider our perspective and proceed with the publication process.
With respect
Best regards,
Dr. Theodoros B. Grivas, MD, PhD
Reviewer 2 Report
Comments and Suggestions for Authors
The manuscript is very insightful and helps the reader understand the criticality of performing scoliosis screening in school children and the rationality for reintroducing the screening program.
Additional comments:
I could come up with some minor improvements that would further strengthen the paper.
1. Additional statistical data could be included to support claims about the global prevalence of scoliosis. Providing averages, trends, or country-specific data would help readers better visualize the scope of the issue.
2. The paper should explore the reasons behind the abandonment of screening programs and the challenges that hinder future implementation. Currently, it only mentions that these programs were discontinued without explaining why.
3. Line 25: ‘this’ programs should be ‘these’ programs. I recommend the authors do a proof reading of the whole manuscript before resubmission.
Author Response
Dear Reviewer 2,
I sincerely appreciate your and other editor’s time and effort in reviewing our article.
We have made revisions in accordance with your suggestions. Regarding the recommendation to improve the English language, the feedback provided was quite general and did not specify particular sections requiring refinement. Nonetheless, we have made several linguistic adjustments.
Regarding the third reviewer's comment that this article should be classified as a review rather than an opinion paper, I fully respect MDPI’s rules and guidelines.
However, I believe that every rule has exceptions, and in this case, reclassifying the article as a review paper would not be appropriate for the following reasons: The argument presented regarding the implementation of the School Scoliosis Sscreening (SSS) is based on 25 years of our own research, which involved extensive data collection over the course of our unique 25-year SSS program. Unlike most similar studies, our research covered a wide age range (5–18 years), allowing us to analyze and uncover significant new insights that had not been previously published. These findings challenge the rationale behind discontinuing SSS in various countries. Restricting the scope of this article by categorizing it as a review article would diminish the strength of our argument. Given the nature of the discussion and the evidence presented, it is more appropriately classified as an opinion paper.
Additionally, the paper’s subheadings on the history and administration of SSS are highly useful in presenting the international policies related to SSS.
I hope you will consider our perspective and proceed with the publication process.
With respect
Best regards,
Dr. Theodoros B. Grivas, MD, PhD
Reviewer 3 Report
Comments and Suggestions for Authors
Dear Editor, I would like to thank you for your kind invitation to review this valuable article submitted to your esteemed journal.
I congratulate the authors for presenting a work that will contribute both to the literature and to professionals working in the field of scoliosis.
I think the article is very valuable, but I have some suggestions for improvement.
I think that the title and the content of the article are not appropriate. In this sense, I suggest that the authors revise the title of the article.
Keywords should be selected using MeSH terms.
I think that the part about menarche in the introduction disrupts the flow of the text.
I suggest adding a reference for Figure 2.
Even if the photograph belongs to the authors, the book or journal in which it is published may be subject to copyright.
If this is an opinion article, it may be valuable for readers if the authors include their own clinical experiences. However, I think it is much more comprehensive than an opinion article. Perhaps the type of article can be changed with the editor's suggestion.
Since the historical development process is described, the references used had to be somewhat dated. This can be considered to be related to the type of the article.
Author Response
Dear Reviewer 3,
I sincerely appreciate your and other editor’s time and effort in reviewing our article.
We have made revisions in accordance with your suggestions. Regarding the recommendation to improve the English language, the feedback provided was quite general and did not specify particular sections requiring refinement. Nonetheless, we have made several linguistic adjustments.
Regarding the third reviewer's comment that this article should be classified as a review rather than an opinion paper, I fully respect MDPI’s rules and guidelines.
However, I believe that every rule has exceptions, and in this case, reclassifying the article as a review paper would not be appropriate for the following reasons: The argument presented regarding the implementation of the School Scoliosis Sscreening (SSS) is based on 25 years of our own research, which involved extensive data collection over the course of our unique 25-year SSS program. Unlike most similar studies, our research covered a wide age range (5–18 years), allowing us to analyze and uncover significant new insights that had not been previously published. These findings challenge the rationale behind discontinuing SSS in various countries. Restricting the scope of this article by categorizing it as a review article would diminish the strength of our argument. Given the nature of the discussion and the evidence presented, it is more appropriately classified as an opinion paper.
Additionally, the paper’s subheadings on the history and administration of SSS are highly useful in presenting the international policies related to SSS.
I hope you will consider our perspective and proceed with the publication process.
With respect
Best regards,
Dr. Theodoros B. Grivas, MD, PhD
Round 2
Reviewer 1 Report
Comments and Suggestions for Authors
The changes that had been made improved the content of the article. The edited form be published.